# Good Scientific Practice and Ethics in Sports and Exercise Science: A Brief and Comprehensive Hands-on Appraisal for Sports Research

**DOI:** 10.3390/sports11020047

**Published:** 2023-02-16

**Authors:** Nitin Kumar Arora, Golo Roehrken, Sarah Crumbach, Ashwin Phatak, Berit K. Labott, André Nicklas, Pamela Wicker, Lars Donath

**Affiliations:** 1Department of Intervention Research in Exercise Training, German Sport University Cologne, 50933 Cologne, Germany; 2Department of Physiotherapy, University of Applied Sciences, 44801 Bochum, Germany; 3Institute of Sport Economics and Sport Management, German Sport University Cologne, 50933 Cologne, Germany; 4Institute of Exercise Training and Sport Informatics, German Sport University Cologne, 50933 Cologne, Germany; 5Institute of Sport Sciences, Otto-von-Guericke University, 39106 Magdeburg, Germany; 6Department of Sports Science, Bielefeld University, 33615 Bielefeld, Germany

**Keywords:** research ethics, research, exercise, review, sport, guidelines

## Abstract

Sports and exercise training research is constantly evolving to maintain, improve, or regain psychophysical, social, and emotional performance. Exercise training research requires a balance between the benefits and the potential risks. There is an inherent risk of scientific misconduct and adverse events in most sports; however, there is a need to minimize it. We aim to provide a comprehensive overview of the clinical and ethical challenges in sports and exercise research. We also enlist solutions to improve method design in clinical trials and provide checklists to minimize the chances of scientific misconduct. At the outset, historical milestones of exercise science literature are summarized. It is followed by details about the currently available regulations that help to reduce the risk of violating good scientific practices. We also outline the unique characteristics of sports-related research with a narrative of the major differences between sports and drug-based trials. An emphasis is then placed on the importance of well-designed studies to improve the interpretability of results and generalizability of the findings. This review finally suggests that sports researchers should comply with the available guidelines to improve the planning and conduct of future research thereby reducing the risk of harm to research participants. The authors suggest creating an oath to prevent malpractice, thereby improving the knowledge standards in sports research. This will also aid in deriving more meaningful implications for future research based on high-quality, ethically sound evidence.

## 1. Introduction

### Historical Milestones of Ethical and Scientific Misconduct in Research

Until the early 19th century, ‘truth’ was fundamentally influenced by cults, religion, and monarchism [1]. With the ‘enlightenment’ of academicians, clinicians and researchers in the 19th century [2], scientific research started to impact the lives of people by providing balanced facts, figures and uncertainties, thereby leading to a better explanation for reality (i.e., evidence vs. eminence). However, dualistic thinking was still interfering with the newer rationalized approach as the estimation of reality by scientific estimation was still being challenged by the dogmatic view of real truth [3].

Over the last decades, researchers underestimated the importance of good ethical conduct [4] in human research by misinterpreting the probabilistic nature of scientific reasoning. Scientific research had constantly been exploited for personal reputations, political power, and terror [3]. The ‘Eugenics program’ originating from the Nazi ideology is an unsettling example of ethical failure and scientific collapse. As part of this program, scientific research was being exploited to justify unwanted sterilization (0.5 million) [5] and mass-killing (0.25 million) [6] for the sake of selection and elimination of ‘unfit genetic material’. In 1955, more than 200,000 children were infected with a Polio vaccine that was not appropriately handled as per the recommended routines [7]. Likewise, the thalidomide disaster of 1962 led to limb deformities and teratogenesis in more than ten thousand newborn children [8]. Considering the aforementioned unethical practices and misconduct, there is a strong need to comply with and re-emphasize the importance of ethics and good scientific practice in humans and other species alike.

In the process of evolution of scientific research, the Nuremberg code laid the foundation for developing ethical biomedical research principles (e.g., the importance of ‘voluntary and informed consent’) [9]. Based on the Nuremberg code and the previously available medical literature, the first ethical principles (i.e., Declaration of Helsinki) were put into practice for safe human experimentation by the World Medical Association in 1964. This declaration proved to be a cornerstone of medical research involving humans and emphasized on considering the health of the patients as the topmost priority [10]. The year 1979 could also be seen as an important milestone, as the ‘Belmont report’ was introduced that supported the idea: ‘the interventions and drugs have to eventually show beneficial effects’. The Belmont report suggests that the recruitment, selection and treatment of participants needs to be equitable. It also highlights the importance of providing a valid rationale for testing procedures to prevent and minimize the risks or harms to the included participants [11].

As a result of the introduction of ethical principles, it became evident that research designs and results should be independent of political influence and reputational gains. There should also be no undeclared conflicts of interest [12]. Interestingly, sports and exercise science emerged as politically meaningful instruments for showing power during the Cold War (i.e., Eastern socialism versus Western capitalism) [13]. Researchers were either being manipulated or sometimes not even published to reduce awareness about the negative effects of performance-enhancing substances [14,15]. Even though these malpractices were strictly against the principles of the Declaration of Helsinki [14], these were prevalent globally, thereby contributing to several incidents of doping in sports [16]. To further minimize unethical research practices, the Good Clinical Practice (GCP) Standards were presented in 1997 to guide the design of clinical trials and formulation of valid research questions [17]. However, some authors criticise the Good Clinical Practice standards as not being morally sufficient to rule out personal conflicts of interest when compared to the ethical standards of the Declaration of Helsinki [18].

Nowadays, professional development and scientific reputation in the research community are related to an increase in the number of publications in high-ranked journals. However, the increasing number of publications gives very little information about the scientific quality of the employed methods, as some of the published papers either contain manipulated results [19] or methods that could not be replicated [12]. Moral and ethical standards are widely followed by sports researchers as evidenced by the applied methods that are mostly safe, justified, valuable, reliable and ethically approved. However, the ethical approval procedures, the dose and the application of exercise training vary greatly between studies and institutions. The review by Kruk et al., 2013 [20] provides a balanced summary of the various principles based on the Nuremberg Code and the Declaration of Helsinki. GCP standards of blinding (subjects and outcome assessors), randomization, and selection are not consistently considered and are sometimes difficult to follow due to limited financial and organizational resources. There is a prevalent trend in the publication of positive results in the scientific community, as negative results often fail to pass editorial review [21]. Additionally, certain unethical research practices have been observed, such as the multiple publication of data from a single trial (referred to as “data slicing”), the submission of duplicate findings to multiple journals, and instances of plagiarism [22]. These limitations negatively affect the power, validity, interpretability and applicability of the available evidence for future research in sports and exercise science. Previous research showed that, if used systematically, lifestyle change and exercise interventions can prove to be one of the most efficient strategies for obtaining positive health outcomes [23] and longevity [24]. Hence, the present article recommends avoiding malpractices and using the underlying ethical standards to balance risks and benefits along with preventing data manipulation and portrayal of false-positive results.

## 2. Codes of Conduct in Sport Research

All the available codes, declarations, statements, and guidelines aim at providing frameworks for conducting ethical research across disciplines. These frameworks generally cover the regulative, punitive, and educational aspects of research. Codes of ethical conduct not only outline the rules and recommendations for conducting research but also outline punishments in case of non-compliance or misconduct. Hence, these ethical codes and guidelines should be considered the most important educational keystones for researchers as these frameworks allow scientists to design and conduct their studies in a better way. Declarations and guidelines are regularly updated to accommodate newer information and corrections. Thus, one also needs to be flexible when using these guidelines as these reflect ongoing scientific and societal development.

Codes and declarations in sport and exercise science regulate both quantitative and qualitative research and include information about human and animal rights, research design and integrity, authorship and plagiarism. We will categorize these guidelines based on the individuals whom guidelines aim to protect (e.g., participants or researchers).

Legal codes and norms of a country are inherently binding to the researchers and institutions who are conducting the research and do not require ratification from the researching individual or organization. These laws can include data storage, child protection, intellectual property rights, or medical regulations applicable to a specific study. However, ethics codes not only cater to the questions of legality but also include moral parameters of research like conducting ‘true’ research. Likewise, if the codes are drafted by a research organization, everyone conducting research for this particular organization is supposed to follow these codes.

Researchers have the responsibility to assess which codes, and standards are relevant to their field of research depending on the country, participants, and research institution (Table 1). This can be confirmed by the academic supervisors or the scientific ethics board of the research institution. While there is a growing number of codes and guidelines for different research fields, it is important to consider that none of these can cater to the needs of every single research design alone. For example, the Code of Ethics of the American Sociological Association (ASA) states: “Most of the Ethical Standards are written broadly in order to apply to sociologists in varied roles, and the application of an Ethical Standard may vary depending on the context” [25]. Hence, as ethical standards are not exhaustive, scientific conduct that is not specifically addressed by this Code of Ethics is not necessarily ethical or unethical [25].

It is crucial to recognize the purpose of an ethics code rather than just following it for ticking boxes. Understanding the aims and limitations of an ethics code will allow for a more meaningful application of the underlying principles to the specific context without ignoring the potential limitations of a study. Unintentional transgressions can occur through subconscious bias, fallacies, or human errors. However, the unintentional errors can be mitigated by following the streamlined process of research conception, method development and study conduct following approval from the Institutional Review Boards (IRBs), Ethical Research Commissions (ERCs), supervisors, and peers. In case of intentional errors, the punitive aspect needs to come into action and the transgressors might need to be investigated and sanctioned, either by the research organizations or by law.

## 3. Differences between Drug and Exercise Trials

Randomized controlled trials (RCTs) are regarded as the highest level of evidence [26,27]. For both the cases (exercise vs. drug studies), RCTs primarily aim at investigating the dose-response relationships and obtaining causal relationships [28]. Drug trials compare one drug to other alternatives (e.g., another drug, a placebo, or a treatment as usual). Likewise, exercise trials often compare one mode of exercise to another exercise or no exercise interventions (e.g., usual care, waitlist control, true control, etc.), ideally under caloric, workload or time-matched conditions. However, placebo or sham trials are still rare in sports and exercise research due to their challenging nature [29]. The following quality requirements should be fulfilled for conducting high-quality exercise trials: (a) ensure blinding of assessors, participants and researchers; (b) placebo/sham intervention (if possible), and (c) adequate randomization and concealed allocation.

### 3.1. Blinding

The term ‘blinding’ (or ‘masking’) involves keeping several involved key persons unaware of the group allocation, the treatment, or the hypothesis of a clinical trial [30,31,32,33,34]. The term blinding and also the types of blinding (single, double, or triple blind) are being increasingly used and accepted by researchers but there is a lack of clarity and consistency in the interpretation of those terms [33,35,36]. Blinding should be conducted for participants, health care providers, coaches, outcome assessors, data analysts, etc. [31,33,34,37]. The blinding process helps in preventing bias due to differential treatment perceptions and expectations of the involved groups [28,30,31,32,38,39,40].

Previous research has shown that trials with inadequately reported methods [41] and non-blinded assessors [42] or participants [43] tend to overestimate the effects of intervention. Hence, blinding serves as an important prerequisite for controlling the methodological quality of a clinical trial, thereby reducing bias in assessed outcomes. Owing to this reason, most of the current methodological quality assessment tools and reporting checklists have dedicated sections for ‘blinding’. For example, three out of eleven items are meant for assessing ‘blinding’ in the PEDro scale [44]; the CONSORT checklist for improving the reporting of RCTs also includes a section on ‘blinding’ [45]. In an ideal trial, all participants involved in the study should be ‘blinded’ [30]. However, choosing whom to ‘blind’ also depends on and varies with the research question, study design and the research field under consideration. In the case of exercise trials, blinding is either not adequately done or poorly reported [36,46]. The lack of reporting might be the result of a lack of awareness of the blinding procedures rather than the poor methodological conduct of the trial itself [34]. Hence, blinding is not sufficiently addressed in exercise, medicine and psychology trials [47,48] due to lacking knowledge, awareness and guidance in these scientific fields leading to an increased risk of bias [48].

Blinding of participants is difficult to achieve and maintain [34,39,40,49] in exercise trials as the participants would usually be aware of whether they are in the exercise group or the control (inactive) group [31,39,50]. Likewise, the therapists are also generally aware of the interventions they are delivering [51], and the assessors are aware of the group allocation because it is common in sports sciences that researchers are involved in different parts of research (recruiting, assessment, allocation, training, data handling analysis) due to limited financial resources. Thus, the adequacy of blinding is usually not assessed as it is often seen as ’impossible’ in exercise trials.

Consequently, we strongly recommend using independent staff for testing, training, control and supervision to improve possibilities of blinding of the individuals involved in the study [39]. Researcher also need to decide if it is methodologically feasible and ethically acceptable to withhold the information about the hypothesis and the study aims [52] from assessors and participants. This needs to be considered, addressed and justified before the trial commences (i.e., a priori). While reporting methods of exercise trials, it is important not only to describe who was blinded but also to elaborate the methods used for blinding [33,48]. This helps the readers and research community to effectively evaluate the level of blinding in the trial under consideration [33,53]. Furthermore, if blinding was carried out, the authors can also include the assessment of success of the blinding procedure [33,54]. Readers can access more information about the various possibilities for blinding using the following link (http://links.lww.com/PHM/A246 accessed on 10 October 2022) [36].

### 3.2. “Placebo” (or Sham Intervention)

‘Placebo’ is an important research instrument used in pharmacology trials to demonstrate the true efficacy of a drug by minimizing therapy expectations of the participants [55]. As the term placebo is generally used in a broad manner, precise definitions are difficult. Placebo is used as a control therapy in clinical trials owing to their comparable appearance to the ‘real’ treatment without the specific therapeutic activity [56]. In an ideal research experiment, it would not be possible to differentiate between a placebo and an intervention treatment [57,58]. The participants should not be aware of the treatment group either, because it can lead to the knowledge of whether they received a placebo or the investigated drug [57]. A review of clinical trials comparing ‘no treatment’ to a ‘placebo treatment’ concluded that the placebo treatment had no significant additional effects overall but may produce relevant clinical effects on an individual level [59]. As outlined previously, the placebo effect is rarely investigated in sports and exercise studies. It is generally investigated using nutritional supplements, ergonomic aids, or various forms of therapy in the few existing studies [60]. Placebos have been shown to have a favorable effect on sports performance research [61], implying that these could be used for improving performance without using any additional performance-enhancing drugs [62].

However, it is quite difficult to have an adequate placebo in exercise intervention studies, as there is currently no standard placebo for structured exercise training [28]. For exercise training interventions, a placebo condition is defined as *“an intervention that was not generally recognized as efficacious, that lacked adequate evidence for efficacy, and that has no direct pharmacological, biochemical, or physical mechanism of action according to the current standard of knowledge”* [63]. As a result, using a placebo in exercise interventions is often seen as impractical and inefficient [57,58]. As the concept of blinding is also linked to the use of a placebo, it is usually difficult to implement in exercise trials.

When it comes to exercise experiments, an active control group is considered to be more effective than a placebo group [10,28]. In other cases, usual care or standard care can also be used as the control intervention [28]. In exercise trials, instead of using the term ‘placebo treatment’, the terms “placebo-like treatment” or “sham interventions” should be applied [64,65]. Previous recommendations by other researchers [61] also underpin our rationale.

### 3.3. Randomization and Allocation Concealment

Group allocation in a research study should be randomized and concealed by an independent researcher to minimize selection bias [66]. Randomization procedures ensure that the differences in treatment outcomes solely occur by chance [28,67]. Several methods for randomization are available; however, methods such as stratified randomization are being increasingly popular as they ensure equal distribution of participants to the different groups based on several important characteristics [66]. Other types of randomization, such as cluster randomization, may be appropriate when investigating larger groups, for example, in multicenter trials [28].

Since researchers are frequently involved in all phases of a trial (recruitment, allocation, assessment and data processing), randomization should usually be conducted by someone who is not familiar with the project’s aims and hypotheses. In studies with a large number of participants, the interaction between subjects and assessors can significantly impact the results [68]. The randomization procedure used in the clinical trial should be presented in scientific articles and project reports so that readers can understand and replicate the process if needed [66]. Based on the aforementioned aspects, exercise trials are not easily comparable to drug trials and the differences lead to difficulties in conducting scientifically conceptualized exercise trials. However, researchers should strive for quality research by using robust methods and providing detailed information on blinding, randomization, choice of control groups, or sham therapies, as appropriate. Researchers should critically evaluate the risk-benefit ratio of exercise so that the positive impacts of exercise on health can be derived and the cardiovascular risks associated with exercise could be minimized [69].

## 4. Key Elements of an Ethical Approval in Exercise Science

As previously described, ethical guidelines are needed to protect study participants from potential study risks and increase the chances of attaining results that ease interpretation. Therefore, a prospective ethical approval process is required prior to the recruitment of the participants [70]. This practice equally benefits the participants by safeguarding them against potential risks and the practitioners who base their clinical decisions on research results. Research results from a study with a strong methodology will enable informed and evidence-based decision making. If the methodology of a research project contains some major flaws, it will negatively affect the practical applicability of the observed results [71]. Various journal reviewers provide suggestions to reject manuscripts without any option to resubmit if no ethical approval information is provided. This demonstrates the importance of ethical approval and proper scientific conduct in research [70].

The following key elements need to be addressed in an ethical review proposal: Introduction, method, participant protection, and appendix. These key elements should be detailed in a proposal with at least three crucial characteristics addressed in each section (Figure 1). This hands-on framework would help to expedite the process of decision-making for members of the ethics committee [72].

The ‘introduction’ section should start with a general overview of the current state of research [4]. Researchers need to describe the rationale of the proposed study in an easy and comprehensible language considering the current state of knowledge on that topic [4]. The description helps to provide a balanced summary of the risks and benefits associated with the interventions in the proposed study. The novelty of the stated research question and the underlying hypothesis must be justified. If the proposed study fails to expand the current literature on the topic under consideration, conducting the study would be a ‘waste’ of time and financial resources for researchers, participants, and funding agencies [73]. Hence, ethical approvals should not be given for research projects that fail to provide novelty in the approach to the respective research area. The introduction should also include information on funding sources including the name of the funding partner, duration of monetary/resource support, and any potential conflicts of interest. If no funding is available, authors should declare that ‘This study received no funding’ [70].

The subsequent ‘methods’ section should include detailed information about the temporal and structural aspects of the study design. Researchers should justify the used study design in a detailed manner [4,28]. Multiple research designs can be utilized for addressing a specific research question, including experimental, quasi-experimental, and single-case trial designs [74,75]. However, a valid rationale should be provided for choosing a randomized cross-over trial design when the gold standard of randomized control trials is also feasible. Readers are advised to refer to the framework laid down by Hecksteden et al., 2018, for extensive information on this section [28]. Researchers should also provide a broad, global and up-to-date literature-based justification for their interventions or methods employed in the study. For instance, if the participants are asked to consume supplements, the recommendations for the dose needs to be explained based on prior high-quality studies and reviews for that supplement [4]. The criteria for subject selection (inclusion and exclusion criteria) and sample size estimation need to be explained in detail to allow replication of the study in the future [76]. Lacking sample size estimations is only acceptable in rare cases and requires detailed explanations (e.g., pilot trials, exploratory trials to formulate a hypothesis, acceptability trials). Moreover, sufficient details should be provided for the measuring devices used in the study and a sound rationale should be provided for the choice of that particular measuring device and the measured parameters [4].

The section on ‘participant protection’ deals with potential risks (physical and psychological adverse outcomes) and benefits to the participants. The focus should be adjusted to the study population under consideration. For example, while conducting a study on a novel weight training protocol with elite athletes, all information and possible effects on the athletes’ performance need to be considered, as their performance level is their ‘human capital’ [4]. The investigators also need to provide information on the individuals responsible for different parts of the study, i.e., treatment provider, outcome assessor, statistician, etc. In some cases, externally qualified personnel are needed during the examination process. For example, a physician might be needed for blood sampling or biopsies and this person should also be familiar with the regulations and procedures to avoid risk to the participants due to a lack of experience in this area. Prior experience and qualifications are required for conducting research with vulnerable groups, such as children, the elderly and pregnant females. Williams et al. (2011) summarized essential aspects of conducting research studies with younger participants [77]. Overall, the personnel should be blinded to the details of the group allocation and participants, if possible [30]. The study applicants also need to provide information about the planned compensation and the follow-up interventions. Harriss and colleagues suggested that the investigators are not expected to offer the treatments in case of injury to the participants during the study (except first aid) [70]. However, this recommendation is not usually documented and translated into research practice.

The ‘appendix’ section should contain relevant details about the following: consent, information to the participants and a declaration of pre-registration. The information to the participant and the consent forms need to be documented in an easy to understand language. A brief summary of the purpose of the study and the tasks to be performed by the participants should also be added. Then, a concise but comprehensive overview of the potential risks and benefits is needed. The next section should include information for participants: the participants’ right to decline participation without any consequence and the right to withdraw their consent at any time without any explanation. The regulations for the storage, sharing and retention of study data need to be detailed [70]. The names and institutional affiliations of all the researchers along with the contact information of the project manager should be listed. A brief overview of the study’s aim, tasks, methods and data acquisition strategies should be described. Finally, consent is needed for processing the recorded personal data [70]. The last section of the ‘appendix’ must include a declaration of pre-registration (e.g., registration in the Open Science Framework or trial registries) to avoid alterations in the procedure afterward and facilitate replication of study methods [78].

## 5. Study Design and Analysis Models

The process of conceptualizing an exercise trial might involve various pitfalls at every stage (hypothesis formulation, study design, methodology, data acquisition, data processing, statistical analysis, presentation and interpretation of results, etc.). Thus, the entire ‘design package’ needs to be considered when constructing an exercise (training) trial [28]. Formulation of an adequate and justified research question is the essential aspect before starting any research study. Formulating a good research question is pivotal to achieve adequate study quality [79]. According to Banerjee et al., 2009 [80], “a strong hypothesis serves the purpose of answering major part of the research question even before the study starts”. As outlined in previous sections, ethical research aspects must be taken into account while framing the research question to protect the privacy and reduce risks to the participants. The confidentiality of data should be ascertained and the participants should be free to withdraw from the study at any time. The authors should also avoid deceptive research practices [79].

Hecksteden et al., 2018, suggested that RCTs can be regarded as the gold standard for investigating the causal relationships in exercise trials [28]. However, it is sometimes not feasible to conduct RCTs in the field of sports science due to logistical issues, such as smaller sample sizes and blinding the location of the study (e.g., schools, colleges, clinics, etc.). In this case, alternative study designs such as cluster-RCTs, randomized crossover trials, N-of-1 trials, uncontrolled/non-randomized trials, and prospective cohort studies can be considered [81]. Considering the complex nature of exercise interventions, the Consensus on Exercise Reporting Template (CERT) has been developed to supplement the reporting and documentation of randomized exercise trials [81]. Adherence to these templates might help to improve the ethical proposal reporting standards when designing new RCTs.

A recent comment, in the journal ‘Nature’, highlights the importance of using the right statistical test and properly interpreting the results. According to the paper, the results of 51 percent of articles published in five peer-reviewed journals were misinterpreted [82]. Frequentist statistics and *p*-values are popular summaries of experimental results but there is a scope for misinterpretation due to the lack of supplementary information with these statistics. For instance, authors tend to draw inferences about the results of a study based on certain ‘*threshold p-values*’ (generally *p* < 0.05) [83]. However, with an increase in sample size, the *p*-value tends to come closer to zero regardless of the effect size of the intervention [83]. With the rise of larger datasets and thus potentially higher sample sizes, the *p*-value threshold becomes questionable. A call for action has recently been raised by more than 800 signatories to retire statistical significance and to stop categorizing results as being statistically significant or non-significant. Recently, researchers suggested using confidence intervals for improving the interpretation of study results [82]. Although alternative methods such as magnitude-based inference (MBI) exists, there is scarce evidence that MBI has checked the use of *p*-value and hypothesis testing by sports researchers [84]. MBI tends to reduce the type II error rate but it increases the type I error rate by about two to six times the rate of standard hypothesis testing [85]. In the next paragraphs, we focus on the commonly used practices within the frequentist statistics domain.

Frequentist statistical tests are categorized into parametric and non-parametric tests. Non-parametric tests do not require the data to be normally distributed, whereas parametric tests do [86]. The following factors help in deciding the appropriate statistical test: (a) type of dependent and independent variables (continuous, discrete, or ordinal); (b) type of distribution, if the groups are independent or matched; (c) levels of observations; and (d) time dependence. Readers can choose the right statistical tests based on the type of research data they are planning to use [87,88]. A recent publication outlined 25 common misinterpretations concerning *p*-values, confidence intervals, power calculations and key considerations while interpreting frequentist statistics [89]. We recommend sports researchers consider the listed warnings while interpreting the results of statistical tests.

Out of the various frequentist statistical methods, analysis of variances (ANOVA) is one of the most widely used tests to analyze the results of RCTs. It does not, however, provide an estimate of the difference between groups, which is usually the most important aspect of an RCT [90]. Linear models (e.g., *t*-tests) suffer from similar issues when analyzing categorical variables, which are a wider part of RCT analysis [91]. Type I errors (false positive, rejecting a null hypothesis that is correct) and Type II errors (false negative, failure to reject a false null hypothesis) are often discussed while interpreting RCT results [80]. Though it is not possible to completely eliminate these errors, there are ways to minimize their likelihood and report the statistics appropriately. The most commonly used methods for minimizing error rates include the following: (a) increasing the sample size; (b) adjusting for covariates and baseline differences [92]; (c) eliminating significance testing; and (d) reporting a confusion matrix [80,86,93].

Mixed logit models are potential solutions for some of the challenges listed above. They combine the advantages of random effects logistic regression analysis with the benefits of regression models [94]. In addition, mixed logit models, as part of the larger framework of generalized linear mixed models, provide a viable alternative for analyzing a wide range of outcomes. For increasing the transparency and interpretability of the observed results, mixed logit classification algorithms and evaluation matrices such as cross validation and presentation of a confusion matrix (type I and type II error rates) can be utilized [86]. Mixed logit models can also be utilized as predictive models rather just ‘inference testing’ models.

## 6. Limitations

Despite extensive efforts to incorporate empirical and current evidence regarding good scientific practice and ethics into this paper, it is possible that some literature may have been omitted. Nonetheless, the paper comprehensively covers key aspects of prevalent ethical misconducts and the standards that should be upheld to prevent such practices. As a result, readers can have confidence in the literature presented, which is based on a substantial body of existing evidence. Readers are also encouraged to engage in critical evaluation and to consider new approaches that could improve the overall scientific literature.

## 7. Conclusions

We highlighted the various pitfalls and misconduct that can take place in sports and exercise research. Individual researchers associated with a research organization need to comply with the highest available standards. They need to maintain an intact ‘moral compass’ that is unaffected by expectations and environmental constraints thereby reducing the likelihood of unethical behavior for the sake of publication quantity, interpretability, applicability and societal trust in evidence-based decision-making. To achieve these objectives, a Health and Exercise Research Oath (HERO) could be developed that minimizes the allurement to cheat and could be used by PhD candidates, senior researchers, and professors. Such an oath would prevent intentional or unintentional malpractices in sport and exercise research, thereby strengthening the knowledge standards based on ethical exercise science research. Overall, this will also improve the applicability and interpretability of research outcomes.

## Figures and Tables

**Figure 1 sports-11-00047-f001:**
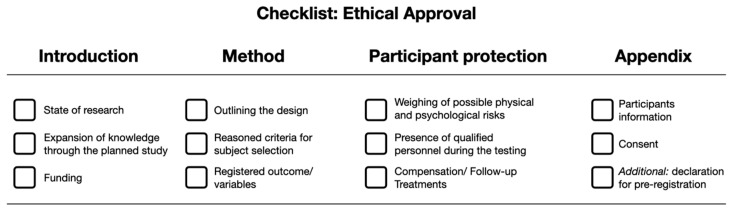
Overview 4 × 3 short list for outlining ethical approval in sport and exercise science.

**Table 1 sports-11-00047-t001:** Detailed overview of Codes, Declarations, Statements and Guidelines relevant for sports and exercise science research.

Whom Does It Protect?	What Are the Topics?	Regulating Declarations, Codes and Guidelines
Research Subject (Humans incl. vulnerable populations, animals, environment)	Anonymity, confidentiality, privacy Informed consent Remuneration Safety and Security Sexual Harassment Gender Identity Human rights Children’s rights Disability rights Animal rights Anti-Doping	WMA Declaration of Helsinki WHO Research Guidelines ASA Code of Ethics APA Ethical Principles NRC Guide BASES Expert Statements UNICEF procedure for ethical standards IOC Medical Code and Consensus Statements WMA Statement on animal use in biomedical research WADA Anti-Doping Code
Research Process	Research questions Study design Data collection Data analysis Result interpretation Result sharing Placebo Randomised Controlled Trials Sample Size Blinding	EQUATOR Reporting Guidelines (CONSORT, etc.) ISA Guidelines UK MRC Guidelines UKRIO Code of Practice MRC Good research practice Montreal Statement on Research Integrity Singapore Statement on Research Integrity EURODAT Guidelines
Researcher (Individuals, Institutions)	Conflicts of Interest Bias Plagiarism Authorship Fraud Governance Transparency Anti-Betting Anti-corruption	University Ethics Codes and Guidelines IOC Charta IOC and IPC Ethics Code AAAS Brussels Declaration

## Data Availability

No new data were created or analyzed in this study. Data sharing is not applicable to this article.

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
