# Peer review of "Good Scientific Practice and Ethics in Sports and Exercise Science: A Brief and Comprehensive Hands-on Appraisal for Sports Research"

_sports, 2023, doi:10.3390/sports11020047_

Round 1
Reviewer 1 Report
Please, see the attached file.

Author Response
We thank the reviewers for sharing their expertise and time. The input has strengthened the manuscript. We highlighted all the changes in the main manuscript and also respond to each specific comment in the attached PDF document.

Reviewer 2 Report
- The overall aim of this study was to review the current literature on good scientific practices in sports and exercise science. The authors placed an emphasis on the idea research in these contexts should be aware of ethical (mis)conducts and how to follow standard procedures accepted internationally. It is important that more studies such as this one be published.
- minor comments:
review citations (e.g., Hecksteden et al., 2018).
- review the manuscript for english changes required.
Author Response
We thank the reviewers for sharing their expertise and time. The input has strengthened the manuscript. We highlighted all the changes in the main manuscript and also respond to each specific comment in the attached PDF.

Reviewer 3 Report
The authors present a good research paper.
- The relevance of the topic: Good.
- Introduction: Can be improved.
- Methodology: Can be improved.
- Results: Good.
- Discussion: Good.
However, ACCEPT AFTER MINOR REVISION. In general, the paper follows an adequate structure and correct scientific support and can be published considering some limitations. The study is interesting in the field of Good Scientific Practice and Ethics in Sports and Exercise Science. However, there are a series of limitations that should be considered.
In the first place, carry out a review of the existing literature related to the subject, being essential to inquire into the MPDI – Sports journal itself, since there are papers related to its manuscript that can help to improve it. Therefore, include those references, if any, especially from the last five years. In addition, recommend reading some papers related to the topic of Good Scientific Practice and Ethics in Sports and Exercise Science:
Harriss, D. J., MacSween, A., & Atkinson, G. (2019). Ethical standards in sport and exercise science research: 2020 update. International Journal of Sports Medicine, 40(13), 813-817.
Kruk, J. (2013). Good scientific practice and ethical principles in scientific research and higher education. Central European Journal of Sport Sciences and Medicine, 1(1), 25-29.
Specific comments.
Title. The title of the manuscript is correct.
Abstract. Incorporate in the summary, a more precise sentence of the results.
Introduction. This section presents the problem in a coherent and clear manner with the correct support of the scientific literature. However, it is convenient to update the references, since there are different documents related to the subject and no mention is made, and it would even be interesting to mention the different existing studies related to Good Scientific Practice and Ethics in Sports and Exercise Science. Also, it could be a future study of review. Some bibliographical references are attached to carry out the section of Good Scientific Practice and Ethics in Sports and Exercise Science:
Caldwell, A. R., Vigotsky, A. D., Tenan, M. S., Radel, R., Mellor, D. T., Kreutzer, A., ... & Boisgontier, M. P. (2020). Moving sport and exercise science forward: a call for the adoption of more transparent research practices. Sports Medicine, 50(3), 449-459.
Navalta, J. W., Stone, W. J., & Lyons, S. (2019). Ethical issues relating to scientific discovery in exercise science. International journal of exercise science, 12(1), 1.
Methods. Modify the method section, and specifically, in the section: Design.
- Study design. To write the design section, we recommend that you take some of the following methodologists as references.
Ato, M., López-García, J. J., & Benavente, A. (2013). A classification system for research designs in psychology. Annals of Psychology, 29(3), 1038-1059.
Montero, I., & León, O.G. (2007). A guide for naming research studies in psychology. International Journal of Clinical and Health Psychology, 7(3), 847-862.
Results. Summary of study data and table are correct.
Discussion. The section Discussion is correct.
Conclusion. Differentiate the discussion of the main conclusions of the study. To do this, you must create this section. And modify the limitations of the study and locate them in said section at the end. Also, they must be direct, and highlight the main contributions of the study.
References. They should be reviewed and updated according to the publication standards. There are many errors in the references. Therefore, correct them and adapt them to the magazine's regulations.
Author Response

(The authors gave the same response as above.)
